# Prediction of High Bell Stages of Necrotizing Enterocolitis Using a Mathematic Formula for Risk Determination

**DOI:** 10.3390/children9050604

**Published:** 2022-04-24

**Authors:** Sonja Diez, Lea Emilia Bell, Julia Moosmann, Christel Weiss, Hanna Müller, Manuel Besendörfer

**Affiliations:** 1Pediatric Surgery, Department for General Surgery, University Hospital Erlangen, Friedrich-Alexander-Universität Erlangen-Nürnberg, 91054 Erlangen, Germany; lea.emilia.bell@fau.de (L.E.B.); manuel.besendoerfer@uk-erlangen.de (M.B.); 2Pediatric Cardiology, University Hospital Erlangen, Friedrich-Alexander-Universität Erlangen-Nürnberg, 91054 Erlangen, Germany; julia.moosmann@uk-erlangen.de; 3Department of Medical Statistics & Biomathematics, Medical Faculty Mannheim, Heidelberg University, 68167 Mannheim, Germany; christel.weiss@medma.uni-heidelberg.de; 4Neonatology and Pediatric Intensive Care, Hospital for Children and Adolescents, University Hospital Marburg, 35033 Marburg, Germany; hanna.mueller@med.uni-marburg.de

**Keywords:** necrotizing enterocolitis, NEC, congenital heart disease, fulminant NEC, clinical score, Bell stage

## Abstract

Necrotizing enterocolitis (NEC) continues to cause high morbidity and mortality. Identifying early predictors for severe NEC is essential to improve therapy and optimize timing for surgical intervention. We present a retrospective study of patients with NEC, treated between 2010 and 2020, trying to identify factors influencing the severity of NEC. Within the study period, 88 affected infants with NEC or NEC-like symptoms are analyzed. A multiple logistic regression analysis reveals the following three independent predictors for NEC in Bell stage III: red blood cell transfusion (*p* = 0.027 with odds ratio (OR) = 3.298), sonographic findings (*p* = 0.037; OR = 6.496 for patients with positive vs. patients without pathological findings) and cardiac anatomy (*p* = 0.015; OR = 1.922 for patients with patent ductus arteriosus (PDA) vs. patients with congenital heart disease (CHD); OR = 5.478/OR = 2.850 for patients with CHD/PDA vs. patients without cardiac disease). Results are summarized in a clinical score for daily application in clinical routine. The score is easy to apply and combines clinically established parameters, helping to determine the timing of surgical intervention.

## 1. Introduction

Severe necrotizing enterocolitis (NEC) is a life-threatening disease and can result in acute sepsis and multiorgan failure. Further, it can cause chronic morbidity due to short bowel syndrome, subsequent liver damage and nutritional deficiencies with dystrophia, neurological and cognitive impairment [1,2]. Despite the improvements in neonatal care, the mortality of NEC remains high, up to 51% in extremely low-birth-weight preterms and 38% in patients with congenital heart disease (CHD) [3,4,5].

The variety of clinical signs that determine the severity and heterogeneity of this disease complicates diagnosis and optimal treatment. Identifying the optimal timing for surgical intervention remains difficult due to the low sensitivity of current diagnostic criteria, leading to inadequate management especially in severe cases. Intestinal perforation is a clear indication for surgical therapy, but, if possible, irreversible intestinal damage should be prevented. Current research is focusing on the development of new biomarkers in the serum [6] and the feces [7] to determine early NEC and scores to classify therapeutic approaches and limit potential sequela. Gephart and colleagues [8] contributed the clinical composite scoring system GutCheckNEC, providing a diagnostic strategy for the early diagnosis of NEC based on a large population-scale study. The score combines nine independent risk factors (gestational age, red blood cell (RBC) transfusion, unit NEC rate, late-onset sepsis, multiple infections, hypotension treated with inotropic medications, Black or Hispanic race, outborn status and metabolic acidosis) and two protective factors (feeding with human milk and probiotics) to predict surgical indications and mortality. Additionally, Markel et al. tried to identify predictive characteristics (neonatal, maternal, radiographic and biochemical factors) within a scoring algorithm to simplify the indication for surgical therapy and the implementation of new treatment modalities [9], such as next-generation probiotics [10] or the use of stem cells and exosomes [11]. However, repeated attempts to identify reliable clinical parameters associated with a severe course of NEC and their clinical application have not yet been successful. New clinical approaches based on new scientific findings in pathogenesis are essential to improve the treatment and outcome of these patients.

The goal of this study is to identify factors with high prognostic and diagnostic accuracy to predict severe clinical courses of NEC.

## 2. Materials and Methods

### 2.1. Patients, Inclusion Criteria and Data Analysis

We present a retrospective study including all infants diagnosed and treated with NEC in our perinatal center level I between January 2010 and December 2020. The study was approved by the local ethics committee in accordance with the declaration of Helsinki (1964) and its later amendments (No 281_19Bc). The local ethics committee did not demand informed consent due to the retrospective character of anonymized data analysis. Data were collected from medical records of all patients. Exclusion criteria were defined as follows: all patients failing to meet modified Bell’s criteria for diagnosis of NEC in low stages [12,13] and all patients additionally diagnosed with volvulus or abdominal wall defects were excluded (see Figure 1). Inclusion criteria were defined as patients with suspected NEC in Bell stage I, with mild to moderate NEC (Bell stage II) or with advanced NEC (Bell stage III). Fulminant NEC was defined retrospectively in all cases with NEC and extensive necrosis, requiring multiple surgical interventions or intestinal resection of >20 cm. After applying in- and exclusion criteria, the identified patients were further classified according to cardiac anatomy. Cardiovascular anatomy was classified as previously described [14]. Neutrophil-to-lymphocyte ratio (NLR), thrombocyte-to-lymphocyte ratio (TLR) and monocyte-to-lymphocyte ratio (MLR) were calculated using the following formulas at the time of indication for surgical therapy of NEC: NLR = absolute neutrophil count/absolute lymphocyte count; TLR = absolute thrombocyte count/absolute lymphocyte count; MLR = absolute monocyte count/absolute lymphocyte count. The indication for abdominal surgery was based on signs of intestinal perforation or the infant’s clinical deterioration despite conservative therapy, including antibiotic treatment and discontinuation of enteral feeds. Indication for surgery was discussed in an interdisciplinary team led by the leading surgeon. The surgical team and decision strategy did not change during the period of the study. According to the two-hit hypothesis of Garzoni et al. [15], intrauterine and peripartal exposure to inflammatory and hypoxic stimuli might be crucial in the development of NEC in preterm infants. We, therefore, summarized potential first hits (chorioamnionitis, peripartal asphyxia, fetal hydrops, etc.) within patient’s characteristics. Additionally, we included RBC and FFP transfusions up until 5 days before diagnosis of NEC.

### 2.2. Imaging Diagnostics

Abdominal ultrasound and abdominal X-ray were performed in all patients with abdominal distension and clinical suspicion of NEC. Abdominal ultrasound was performed by experienced sonographers. Interpretation of abdominal X-ray was performed by senior pediatric radiologists and all diagnostic findings were reviewed by the same senior radiologist during the study’s period. Sonographic diagnostics were conducted with SIEMENS Acuson X and NX series in its latest generation and optimized settings for neonatal abdominal ultrasound.

Pathological sonographic findings in the abdominal ultrasound included signs of intestinal inflammation (such as hyperperfusion, thickening of the intestinal wall or the mesenterium, destruction of the layers of the intestinal wall and presence of free fluid in the peritoneal cavity), intramural gas (pneumatosis intestinalis) and/or portal venous gas (pneumatosis hepatis) with portal venous gas [16]. Pathological findings in the radiological diagnostics included pneumatosis intestinalis and/or pneumatosis hepatis as well as direct signs of intestinal perforation (pneumoperitoneum) [17].

### 2.3. Statistical Analysis

Data were recorded as quantitative variables or categorial factors. For quantitative or ordinally scaled data, median value together with minimum and maximum were given. For qualitative factors, absolute and relative frequencies were assessed. SAS software (release 9.4; SAS Institute Inc., Cary, NC, USA) was used for statistical analyses. Comparisons between the two study groups were conducted using Mann-Whitney U test for quantitative variables or Chi-square test for qualitative factors. If the preconditions of the Chi-square test were not fulfilled, Fisher’s exact test was applied instead. In cases of comparison of three groups (i.e., ventilation before NEC with 3 categories) and significant test results, subsequent pairwise comparisons were performed using Bonferroni correction. In order to investigate the association between severe NEC cases and clinical variables, multiple logistic regression analysis was performed with different target variables (fulminant NEC, Bell stage III), including all relevant clinical parameters as independent variables using “forward selection” option. For each logistical regression analysis, the area under the curve (AUC) was assessed in order to quantify the goodness of the statistical model. In general, test results with *p*-values smaller than 0.05 were considered as statistically significant.

## 3. Results

### 3.1. Baseline Clinical Variables of Patients

In total, 131 patients with NEC or NEC-like symptoms were diagnosed and treated within the study’s period at the University hospital. After applying in- and exclusion criteria, a total of 88 patients was identified for further analyses (Figure 1). Demographic characteristics of all patients are illustrated in Table 1. The median gestational age at birth was 33.1 weeks (postmenstrual age), with a median birth weight of 1745 g (range: 490–4120 g). The median age at NEC onset was 13 days of life (range: 0–94 days). The survival rate within the study population was 91% (deaths in 8/88 patients). CHD was diagnosed in 29 patients with 74% of patients presenting with a cyanotic cardiac heart defect (21/29 patients with CHD). In total, 41% of CHD patients (n = 12) developed NEC prior to cardiac surgery. Surgical strategies included open abdominal exploration in all surgical cases and additional intestinal resection if needed. In all cases, ostomies were conducted and primary anastomoses were avoided due to inflammation. Primary peritoneal drainage was not performed in this study population. Fulminant NEC was seen in 13 patients (15%). In 62% of infants with fulminant NEC, postoperative short bowel syndrome led to long-term parenteral nutrition (8 out of 13 patients). Distribution of Bell stages included 22 infants with Bell stage I (25%), 35 infants with Bell stage II (40%) and 31 infants with Bell stage III (35%). Patients with Bell stage III were analyzed separately, and comparative data to lower Bell stages are presented in Table 2. Prenatal risk factors were evaluated but failed to reach statistical significance. Additionally, inflammatory markers were tested for association to high Bell stages, in which no significant association could be observed. 82/88 included patients received an abdominal ultrasound (93%) and 62 patients were diagnosed with an additional X-ray (70%). Figure 2 illustrates the differences in diagnostic results of sonographic and radiological imaging, classified by Bell stages. Intramural or portal venous gas was identified in 89% of patients in Bell stage II (n = 31/35) and 65% of patients in Bell stage III (n = 20/31), whereas radiography could only confirm these signs in 9/35 patients of Bell stage II (26%) and 7/31 patients of Bell stage III (23%).

### 3.2. Multifactorial Analysis of High Bell Stages

The multiple regression analysis revealed RBC transfusion (*p* = 0.027, odds ratio (OR) = 3.298), sonographic findings (*p* = 0.037; OR = 6.496) and cardiac anatomy (*p* = 0.015) as predictive factors for Bell stage III. The risk for Bell stage III was higher in patients with CHD or patent ductus arteriosus (PDA) compared to patients without cardiac disease (OR = 5.478 and OR = 2.850, respectively). Comparing patients with CHD to patients with PDA resulted in an OR of 1.922.

This resulted in the following mathematical equation, estimating the risk for the development of NEC diagnosed at Bell stage III:$$\mathrm{Probability}\;\mathrm{of}\;\mathrm{NEC}\;\mathrm{Bell}\;\mathrm{stage}\;\mathrm{III}=\frac{e^{\left(C+1.1934\cdot R+1.8713\cdot S\right)}}{1+e^{\left(C+1.1934\cdot R+1.8713\cdot S\right)}}\;$$
where C=−1.9409, C=−2.5942 or C=−3.6416 for patients with CHD, with PDA or without cardiac disease, respectively. R=1 for patients with RBC transfusion and R=0 otherwise. S=1 in case of pathological findings in ultrasound diagnosis, S=0 otherwise.

CHD was associated with an elevated risk for the development of Bell stage III in comparison to patients with PDA or preterm patients without cardiac malformations. This was illustrated by an OR higher than 1 (OR = 1.922 and OR = 5.478, respectively). Accordingly, RBC transfusion (OR = 3.298) and pathological sonographic findings in the abdominal ultrasound (OR = 6.496) were associated with an increased risk for Bell stage III. The AUC of this multiple model was 0.776.

Based on this model, we were able to present a risk score for clinical daily practice, reflecting the interaction of pathological sonographic findings (one point), RBC transfusion (one point) and cardiac disease (two points; see Figure 3). Hence, this score ranged between 0 (patients without cardiac disease, without pathological findings and without RBC transfusion) and 4 points. The AUC of the corresponding statistical model was 0.768 (*p* = 0.0001). The OR of 2.408 indicated that the risk for Bell stage III increased 2.4-fold with each score point. Using a cut-off value of three (indicating that only patients with three or four points would develop NEC diagnosed at Bell stage III) led to a sensitivity of 68% and a specificity of 69% of the score. When tested within our cohort, 19 out of 28 patients with Bell stage IIII corresponded to a score value of 3 or 4, whereas 38 out of 55 patients with lower Bell stages had a score value between 0 and 2.

We evaluated the influence of the cardiac anatomy and pathological findings on ultrasound or the need of RBC transfusion based on the different pathogenesis of NEC in preterm infants and infants with CHD within this model. Due to the small patient population, we compared proportional differences within the subgroups Bell stage III and Bell stage I-II. Results are shown in Table 3 and excluded a substantial influence of the cardiac anatomy in both subgroups.

Furthermore, we repeated the multiple regression analysis and created two groups: patients with CHD and preterm patients with/without PDA, as the study’s sample size was too small to compare three groups. We were able to see that this model resulted in the same predictors for Bell stage III: pathological findings in the sonographic diagnostics and RBC transfusion.
$$\mathrm{Probability}\;\mathrm{of}\;\mathrm{NEC}\;\mathrm{Bell}\;\mathrm{stage}\;\mathrm{III}=\frac{e^{\left(C+1.9690\cdot S+1.2915\cdot R\right)}}{1+e^{\left(C+1.9690\cdot S+1.2915\cdot R\right)}}$$
where C=−2.0633 or C=−3.5151 for patients with CHD or without cardiac disease, including PDA patients, respectively. R=1 for patients with RBC transfusion and R=0 otherwise. S=1 in the case of pathological findings in ultrasound diagnosis, S=0 otherwise.

### 3.3. Analysis of Fulminant NEC

The target variable “fulminant NEC” was additionally analyzed with a multivariable regression analysis. However, only one factor was revealed to be significant: therapy with fresh frozen plasma (FFP) was significantly associated with fulminant NEC (OR = 7.536, *p*-value = 0.004 with AUC = 0.731). A patient receiving treatment with FFP showed a 30% increased probability to develop fulminant NEC as opposed to 5.5% in cases without the substitution of FFP.

## 4. Discussion

The identification of risk factors predicting patients at risk for severe NEC could improve the outcome for this vulnerable patient population [18]. The aim of this study was to identify variables predictive of NEC cases in high Bell stages. We identified CHD, pathological findings in abdominal ultrasound and RBC transfusion as independent risk factors in the development of NEC in Bell stage III. This was summarized in a mathematical model and clinical score to predict high Bell stages. Upperman et al. published an alternative mathematical strategy with the goal of better understanding complex inflammatory diseases, but could not find a clinical reference [19].

The identified variables have all been discussed in recent literature, but their influence on the severity or clinical course of NEC remains controversial, as discussed below.

There is consensus on the pathogenesis of NEC with a distinction between cardiac and inflammatory NEC, which has been described by various authors within large population-based studies and is based on differences in incidence and mortality [20]. Neu et al. even proposed to diagnose an ischemic intestinal necrosis consistently apart from the typical NEC [21]. Characteristic clinical findings, such as a predominant colic localization [22] or elevated levels of leucocytes and neutrophils [23], were identified in association with cardiogenic NEC. Cyanotic heart diseases caused impaired intestinal perfusion and reduced intestinal arterial oxygen saturation. A PDA led to left-to-right shunting, resulting in pulmonary overcirculation and diastolic “runoff” in the aorta, contributing to organ hypoperfusion and hypoxia. Changes in oxygen delivery to the gut in the presence of a PDA might initiate the ongoing inflammatory process of severe NEC [24]. We identified CHD as an independent risk factor for higher Bell stages and reduced outcome.

Several studies suggested an influence of RBC transfusion and the development of NEC [25], based on the hypothesis that severe anemia or transfusions might evoke a vasoconstrictive stimulus leading to a dysfunction in the epithelial barrier [26]. Furthermore, replacing fetal hemoglobin with adult hemoglobin showing different oxygen binding characteristics and increased O_2_ radicals is discussed as a potential cause or risk factor for various neonatal diseases [27]. The background of this pathophysiological process remains elusive, as patients with severe NEC might develop bleeding and anemia and, therefore, require RBC transfusions. Moreover, larger meta-analyses were not able to confirm this association [28]. We included RBC transfusion up until 5 days before diagnosis of NEC in our analysis, and we confirmed that RBC transfusions were independently associated with Bell stage III, but did not further investigate the underlying processes. Anemia with aggravating hypoxia might be a possible explanation why infants who require RBC transfusion have a higher risk for severe NEC. Similar might be the influence of FFP transfusion on fulminant NEC within the presented study. According to our data, FFP application is an independent risk factor for a fulminant course of NEC. To our knowledge, there is no described association between FFP transfusion and the occurrence of NEC. A possible explanation could be that severe inflammation and sepsis was accompanied by a disseminated intravascular coagulation requiring FFP or other coagulation products.

Our results emphasized the role of abdominal ultrasound in the diagnosis and monitoring of treatment effects in NEC. It is known to enable a timely diagnosis and a reliable monitoring without radiation exposure. Further advantages include the real-time assessment of intramural and portal venous gas, thickening of the intestinal wall, hypoperfusion and hypoperistalsis [25,29,30]. However, it still does not play a superior role in NEC diagnostics due to suggested reasons as its low sensitivity and negative predictive value [16]. Abdominal radiography remains the gold standard imaging modality in the majority of neonatological centers, even if only late signs of progressed NEC are visible [30]. Based on the results of our study, we see great advantages in abdominal ultrasound and a certain superiority, if performed by an experienced sonographer. Our results highlight its role and support the recent publication of Alexander et al., who stated that pneumatosis and portal venous gas are associated with NEC but are not specific for surgery [16]. This diagnostic approach might substantially improve the diagnosis and, subsequently, the outcome of NEC, especially if validated biomarkers can be added in the future.

The identification of noninvasive variables with good sensitivity and specificity for early diagnosis is crucial to reduce irreversible intestinal damage and improve outcome for patients with NEC [31]. Research on biomarkers has been intensified [32], but still falls short in clinical application. The advantage of our study is that we included well-known variables in NEC of high Bell stages, such as cardiac anatomy and RBC transfusion, and combined them to support the decision between conservative therapy and timely surgical intervention. Furthermore, the importance of ultrasound diagnostics in NEC as a sensitive and specific diagnostic approach was sought to be highlighted. The summarizing score is easy to apply in daily routine and might serve as a baseline estimation for early surgical intervention when positive (value > 3).

This study had several limitations: the single center character with a moderate sample size limited the ability to draw strong conclusions. Due to the retrospective character of the study and the important improvements in neonatal care within the study’s period, conclusions might furthermore be limited. It especially impeded the ability to solve potential biases, such as, e.g., changing the primary examiner and radiologists and ultrasound machines with different quality of imaging diagnostics. Although the sample size did not allow to test three different groups (CHD versus preterm infants with PDA versus preterm infant without PDA), the summary of different multivariate tests enabled the determination of predictor sonographic findings and RBC transfusion and emphasized the role of cardiac defects or the presence of PDA during NEC. We suggest confirming and evaluating the mathematical score prospectively in a new cohort to prove its reliability. Additionally, the sensitivity and specify of the score could be improved further.

## 5. Conclusions

An improved diagnostic algorithm to identify severe NEC cases with a timely surgical indication is mandatory to reach high standards of current neonatal care and elevate the survival of this disease. The score presented in this study is easy to apply in daily clinical routine. Larger population studies in different regions and countries are needed to fill in these essential gaps of knowledge in NEC management and to support the practicability of this mathematic formula for risk determination.

## Figures and Tables

**Figure 1 children-09-00604-f001:**
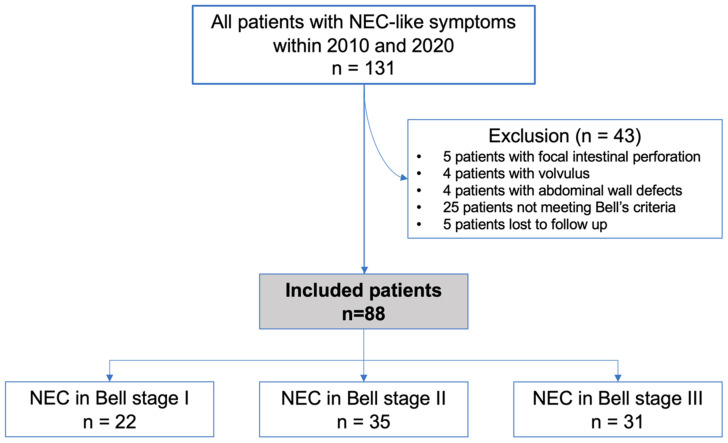
Flow diagram of the selection process of the study’s population.

**Figure 2 children-09-00604-f002:**
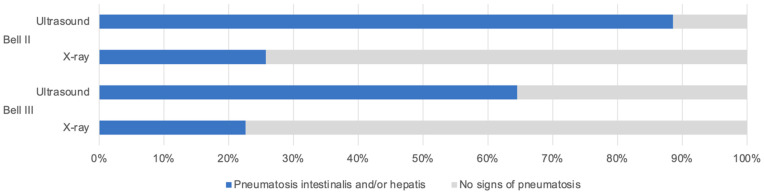
Pneumatosis intestinalis and/or hepatis. Illustrated are patients in Bell stage II and III, diagnosed with intramural or portal venous gas on ultrasound and X-ray.

**Figure 3 children-09-00604-f003:**
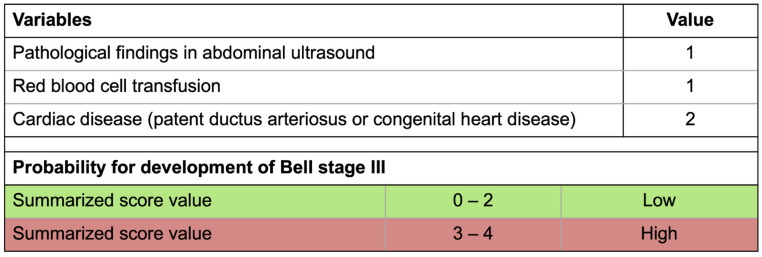
Bell stage III score. Independent predictors for NEC in Bell stage III according to the presented equation summarized in a score for clinical application. Presence of each factor increases risk for high Bell stage 2.4-fold.

**Table 1 children-09-00604-t001:** Basic demographic and clinical data. Summary of patients’ characteristics.

Parameter	Study Population (n = 88)
Sex (n (%))	
Male	57 (65%)
Female	31 (35%)
Umbilical cord pH (median (range))	7.3 (6.7–7.5)
Gestational age at birth (weeks; median (range))	33.1 (23.0–41.6)
Weight at birth (g; median (range))	1745 (490–4120)
Ventilation before NEC (n (%))	
Intubation/ventilation	61 (69%)
CPAP	12 (14%)
None	15 (17%)
First hit (pre- and perinatal inflammatory and/or hypoxic stimuli) (n (%))	
Yes	10 (11%)
No	78 (89%)
Maternal risk factors (chorioamnionitis, preeclampsia) (n (%))	
Yes	4 (4%)
No	84 (96%)
Cardiac status (n (%))	
CHD	29 (33%)
PDA	12 (14%)
No cardiac malformation	47 (53%)
Surgical treatment of cardiac malformation (CHD, PDA) before NEC (n (%))	
Yes	31 (35%)
No	57 (65%)
Prostaglandin application (n (%))	
Yes	21 (24%)
No	67 (76%)
Application of COX-2-inhibitors (n (%))	
Yes	14 (16%)
No	74 (84%)
Transfusion of red blood cells (n (%))	
Yes	37 (42%)
No	51 (58%)
Transfusion of fresh frozen plasma (n (%))	
Yes	33 (38%)
No	55 (62%)
Transfusion of platelets (n (%))	
Yes	23 (26%)
No	65 (74%)
Application of inotropes (n (%))	
Yes	38 (43%)
No	50 (57%)
Nutrition before diagnosis (n (%))	
None	15 (17%)
Human milk	9 (10%)
Formula	19 (22%)
Both	45 (51%)
Age at NEC diagnosis (days; median (range))	13.0 (0–94)
Bell stage (n (%))	
Ia	9 (10%)
Ib	13 (15%)
IIa	19 (22%)
IIb	16 (18%)
IIIa	4 (4%)
IIIb	27 (31%)
Treatment of NEC (n (%))	
Surgery	40 (45%)
Conservative treatment	48 (55%)
Number of needed surgeries (n (%))	n = 40
1	28 (70%)
2 or more	12 (30%)
Proof of intraoperative intraperitoneal bacteria (n (%))	n = 40
Yes	18 (45%)
No	22 (55%)
Macroscopic intestinal necrosis (n (%))	n = 40
Yes	31 (78%)
No	9 (22%)
Macroscopic intestinal perforation (n (%))	n = 40
Yes	27 (67%)
No	13 (33%)
Short bowel syndrome (n (%))	
Yes	8 (9%)
No	80 (91%)
Outcome (n (%))	
Survival	80 (91%)
Death	8 (9%)
C-reactive protein at diagnosis (mg/L; median (range))	7.0 (0–288.7)
Neutrophil-to-lymphocyte-ratio at diagnosis (median (range))	1.7 (0.11–42.11)
Monocyte-to-lymphocyte-ratio at diagnosis (median (range))	0.3 (0–4.95)
Thrombocyte-to-lymphocyte-ratio at diagnosis (median (range))	121.6 (8.5–1275)
Maximum C-reactive protein (mg/L; median (range))	40.9 (0–325.2)
Maximum neutrophil-to-lymphocyte-ratio (median (range))	1.6 (0.11–42.11)
Maximum monocyte-to-lymphocyte-ratio (median (range))	0.3 (0–4.95)
Maximum thrombocyte-to-lymphocyte-ratio (median (range))	89.5 (19.7–1580)

Abbreviations: NEC: necrotizing enterocolitis; PDA: patent ductus arteriosus; CHD: congenital heart defect.

**Table 2 children-09-00604-t002:** Clinical parameters of patients with Bell stage III in comparison to patients with NEC in Bell stage I–II. Significant findings were highlighted.

	Bell Stage I–II (n = 57)	Bell Stage III (n = 31)	*p*-Value	Test
Umbilical cord pH (median (range))	7.3 (7.0–7.5)	7.3 (6.7–7.4)	0.4360	U test
Gestational age at birth (weeks; median (range))	33.4 (23.0–41.6)	32.7 (23.6–39.6)	0.9886	U test
Weight at birth (g; median (range))	1680 (530–4120)	1870 (490–3680)	0.9289	U test
Ventilation before NEC (n (%))			0.0039 *	Chi^2^
Intubation/Ventilation	33 (58%)	28 (90%)		
CPAP	12 (21%)	0		
None	12 (21%)	3 (10%)		
First hit (pre- and perinatal inflammatory and/or hypoxic stimuli) (n (%))			0.7367	Fisher
Yes	6 (11%)	4 (13%)		
No	51 (89%)	27 (87%)		
Cardiac status (n (%))			0.0135 **	Chi^2^
CHD	14 (25%)	15 (49%)		
PDA	6 (10%)	6 (19%)		
No cardiac malformation	37 (65%)	10 (32%)		
Application of catecholamines (n (%))			0.0114	Chi^2^
Yes	19 (33%)	19 (61%)		
No	38 (67%)	12 (39%)		
Nutrition before diagnosis (n (%))			0.2661	Chi^2^
None	7 (12%)	8 (26%)		
Human milk	5 (9%)	4 (13%)		
Formula	12 (21%)	7 (22%)		
Both	33 (58%)	12 (39%)		
Age at diagnosis (days; median (range))	10.0 (0–49)	20.0 (0–94)	0.2112	U test
Positive findings on abdominal ultrasound (n (%))			0.0455	Chi^2^
Yes	41 (75%)	29 (93%)		
No	16 (25%)	2 (7%)		
Intraoperative proof of intraperitoneal bacteria (n (%))	n = 10	n = 30	0.4645	Fisher
Yes	3 (30%)	15 (50%)		
No	7 (70%)	15 (50%)		
Macroscopic intestinal perforation (n (%))	n = 10	n = 30	<0.0001	Fisher
Yes	1 (10%)	26 (87%)		
No	9 (90%)	4 (13%)		
Maximum C-reactive protein (mg/L; median (range))	7.5 (0–259.6)	167.2 (1.5–325.2)	<0.0001	U test

Abbreviations: NEC: necrotizing enterocolitis; PDA: patent ductus arteriosus; CHD: congenital heart defect. * Intubation vs. CPAP: *p* = 0.0069; intubation vs. none: *p* = 0.2022; CPAP vs. none: *p* = 0.6924. ** CHD vs. PDA: *p* = 1; CHD vs. no cardiac malformation: *p* = 0.0183; PDA vs. no cardiac malformation: *p* = 0.2061.

**Table 3 children-09-00604-t003:** Exclusion of confounding parameters. Cardiac status was not a substantially influencing variable of the score, as shown with comparable delta in both subgroups.

	Pathological Findings in Ultrasound Diagnosis			RBC Transfusion		
	Bell Stage I–II	Bell Stage III	Delta	Bell Stage I–II	Bell Stage III	Delta
CHD	64%	87%	23%	36%	60%	24%
PT/PDA	78%	100%	25%	30%	63%	23%

Abbreviations: CHD: congenital heart defect; PT: preterm infants; PDA: patent ductus arteriosus.

## Data Availability

Not applicable.

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
