# Peer review of "Prediction of High Bell Stages of Necrotizing Enterocolitis Using a Mathematic Formula for Risk Determination"

_children, 2022, doi:10.3390/children9050604_

Round 1

Reviewer 1 Report

This study addresses a relevant issue for clinical neonatologists, as the early prediction and prompt diagnosis of NEC remains challenging despite the increasing amount of studies on the topic. However, some inappropriatenesses should be addressed:

  • In the Introduction, the world "inaccuracy" at line 40 should be replaced; the issue of most biomarkers is their lack of sensitivity (f.i. US markers)
  • In material and methods the inclusion criteria, in addition to the exclusion criteria, should be specified. Furthermore, the authors include even the suspected NEC cases (Bell's stage I); fulminant NEC is defined as rapidly progressing NEC in most of the studies and eventually culminating in extensive necrosis or NEC: was this the same definition?
  • Among the materials and methods, criteria for surgery should be better specified, also because between 2010 and 2020 some local practices may have changed (if this not the case, this should be stated)
  • In the results, a separate analysis for patients with CHD should be performed bacause, as correctly written by the authors, the pathogenensis of classical and cardiac NEC is probably different
  • In Figure 1, patients were not randomized as this is a retrospective study
  • What type of transfusions were considered? the ones within 48 or 72 hours from NEC diagnosis or the ones occurred from birth? Cautios should be taken when drawing conclusion at this regard, as the most sick infants requiring many transfusions from birth may also be the ones at higher risk of hypoperfusion and NEC
  • Positive findings at sonography are per definition found only from Bell's stage II and above (therefore the significant p in Table 2 is quite obvious); the results however, as designed in Figure 2, seem to support that pneumatosis and portal gas are associated with NEC but not specific for surgery (Alexander et al. Arch Dis Child Fetal Neonatal Ed 2020)
  • In the Discussion, the authors should highlight among the limitations the retrospective nature of the study;
  • Future plan for confirming and evaluating the mathematical score prospectively in a new cohort might be an idea

Reviewer 2 Report

This is a retrospective study on the prediction of severe NEC. In the current literature, few information is available regarding pre-natal and post-natal factors predisposing severe NEC. As a result, it might be useful have more information about this topic in order to create simple predictive tools to predict the development of severe NEC in clinical practice. I think that it is very interesting study. However I believe that there are some gaps: 

  • Few maternal data are considered. How about preeclampsia, clinical and histological chorioamnionitis, Prenatal doppler? In our experience preeclampsia and histological chorioamnionitis were risk factors for the development of severe NEC.We suggest to add these information if data are available. 
  • Why did you not separate perinatal asphyxia and potential maternal risk factors? 
  • Few post-natal data are considered. How about Leukocitosis, Inflammatory markers at the timing of NEC diagnosis? Were are differences among the NEC groups based on their severity? 
  • In the result section line 122-123 please specify the type of surgical approach. Did you perform a stoma or did you do a resection and primary anastomosis?
  • I suggest to try to do another analysis separating Cardiogenic NEC and other NEC as the pathogenesis might be slightly different. Might the predictive model differ in these two groups?

Reviewer 3 Report

Necrotizing enterocolitis (NEC) remains one of the major challenges of neonatologist, and it has important consequences for preterm newborns, as sepsis or death. Diez et al. performed a retrospective study to identify factors with high prognostic and diagnostic accuracy to predict severe clinical courses of NEC, elaborating a clinical score for NEC severe development. I admired the study and the clinical utility of this, however to my opinion there are some criticisms that should be resolved before the publication:

Major comments:

  1. Lines 43-45, references are missing (I suggest https://doi.org/10.1097/mpg.0000000000001588). In addition, I suggest to describe not only serum, but also fecal biomarkers for early NEC diagnosis, as HMGB1 (https://doi.org/10.3389/fped.2021.672131);
  2. Line 62. I suggest to remove “and to pay special attention to cardiac malformations in NEC patients”, because there is no mention before in the introduction section. Otherwise, you should cite something about the relation between cardiac malformation and NEC;
  3. Line 75. Reference regarding the modified Bell’s criteria are needed;
  4. In methods section, you must cite the types of machines used for image scans, are they the same during the 10 years of the study? The sonographers and radiologists are more than 1? This could be potential bias for the study, that should be described in limitations section;
  5. Lines 91-97. References regarding pathological findings of NEC are needed, for both sonography and radiology;
  6. Which test did you used for the calculation of p-values of the Table 2? For example, in “ventilation”, did you compare the 2 study groups of NEC stages? In this case should be presented 3 p-values, please better describe. In addition, I suggest to improve statistical section, that appears not clear;
  7. In an interesting sub analysis including the cases of “fulminant NEC” (How many?), you specified that fresh frozen plasma therapy is an independent risk factor for fulminant NEC, why did you not include this factor in the clinical score?;
  8. Limitations section should be improved. Besides the fact that there is a small sample size, you should underline that this is a retrospective study, and in 10 years the clinical management of the newborns could been different thanks to the improvement of medical support. In addition, please add the limitations that I suggested in the previous points (see point 4).

Minor comments:

  1. The abbreviations in the tables should be added in notes after the tables not in title, as journal guidelines require;
  2. English should be improved.

Round 2

Reviewer 1 Report

Thank you for addressing all the points.

Reviewer 3 Report

Congratulations !